# Using the Concept of Health Literacy to Understand How People Living with Motor Neurone Disease and Carers Engage in Healthcare: A Longitudinal Qualitative Study

**DOI:** 10.3390/healthcare10081371

**Published:** 2022-07-24

**Authors:** Camille Paynter, Susan Mathers, Heidi Gregory, Adam P. Vogel, Madeline Cruice

**Affiliations:** 1Department of Audiology and Speech Pathology, University of Melbourne, 550 Swanston Street, Melbourne, VIC 3010, Australia; vogela@unimelb.edu.au; 2Calvary Health Care Bethlehem, 152 Como Parade West, Parkdale, VIC 3195, Australia; susan.mathers@calvarycare.org.au (S.M.); heidi.gregory@monash.edu.au (H.G.); 3School of Clinical Sciences, Monash University, 246 Clayton Road, Clayton, VIC 3168, Australia; 4Eastern Health Clinical School, Monash University, 5 Arnold Street, Box Hill, VIC 3128, Australia; 5Redenlab, 585 Little Collins Street, Melbourne, VIC 3000, Australia; 6School of Health Sciences, City, University of London, Northamptom Square, London EC1V 0HB, UK; m.cruice@city.ac.uk

**Keywords:** amyotrophic lateral sclerosis, motor neurone disease, health literacy, longitudinal qualitative research

## Abstract

The growing body of information-seeking and decision-making literature in motor neurone disease (MND) has not yet explored the impact of health literacy. Health literacy relates to the skills people have to access, understand, and use health information and is influenced by motivation to engage with healthcare. We aimed to better understand how people affected by MND engage in healthcare by examining longitudinal interview data using the construct of health literacy. Semi-structured interviews were conducted with 19 persons living with MND and 15 carers recruited from a specialist MND clinic using maximum variation sampling. Transcripts were deductively coded using a framework of health literacy behaviours. The analysis used a matrix-based approach for thematic analysis of longitudinal data. People living with MND and carers sought nuanced information dependent on their priorities and attitudes. Information uptake was influenced by perceived relevancy and changed over time. Time allowed opportunity to reflect on and understand the significance of information provided. The findings indicate that persons living with MND and carers benefit when information and consultations are adapted to meet their communication needs. The results highlight the potential benefits of gaining an early understanding of and accommodating the communication needs, personal preferences, and emotional readiness for information for persons living with MND and their carers.

## 1. Introduction

Motor neurone disease (MND) or amyotrophic lateral sclerosis (ALS) is an adult-onset progressive neurological disease characterised by insidious weakness of voluntary muscles, resulting in paralysis (muscle weakness), dysarthria (speech difficulty), dysphagia (swallowing difficulty), and dyspnoea (breathing difficulty) [1]. Disease onset, symptoms, and progression is variable and not all patients will experience all of these symptoms [1]. Up to 50% of people may develop cognitive and/or behavioural changes during the disease course [1]. There is no curative treatment for MND; therefore, clinical management focuses on symptom management and quality of life. There are two disease-modifying therapies: riluzole, which shows a small survival benefit and is widely used, and edaravone, which shows disease slowing in highly selected patients and has limited approval for use worldwide [1]. The progressive nature of the disease necessitates ongoing, often time-dependent decision making regarding non-invasive or invasive ventilation (support for dyspnoea), gastrostomy (feeding tube) insertion, alternative and augmentative communication aids (AAC), and future care planning [2]. Non-invasive ventilation (NIV) is a type of assisted ventilation for patients with dyspnoea that is shown to improve quality of life and survival [3]. Gastrostomy may be suggested to people living with MND when they are considering NIV and/or experiencing dysphagia as a method for providing nutritional support, stabilising weight loss, and administering medication [4]. For a proportion of people living with MND, gastrostomy improves quality of life [5]; however, its impact on survival remains unclear [4]. Coordinated clinical care provided by multidisciplinary team clinics is shown to enhance survival compared with patients managed solely by general neurology clinics [2].

The clinical management of MND involves information provision, intervention recommendations, symptom management, and advance care planning. Information provision needs to be person-specific since the information preferences of persons living with MND and carers can vary in timing, depth, and topics [6]. Understanding patient and carer preferences for information and healthcare engagement may reduce the emotional cost of receiving information about disease course, prognosis, and early or unexpected interventions [7,8]. Patient–clinician relationships can be strengthened during the process of negotiating engagement, although instruments to gauge patient preference for engagement in healthcare vary in quality [9]. Persons living with MND prefer information about disease symptoms, course and prognosis, and potential research options [10,11,12]. Whilst carers seek similar information from persons living with MND, they also require information on care provision, community services, assistive devices, and managing future functional decline [12,13,14]. Carers may also seek information earlier in the disease course than persons living with MND [6].

The way that people interact with healthcare information is known as health literacy. Traditionally described in the fields of public health and health promotion, health literacy is defined as the skills required to access, understand, and use information to make decisions about health. More contemporary views within healthcare widen this definition and recognise it is a multidimensional construct involving contextual factors, cognitive and social skills, social support, and personal motivation to engage with healthcare information [15,16], which is shaped by relationships with healthcare professionals (HCP) and the health system at large [17]. Health literacy in the MND community has not been as widely explored as it has in other health conditions such as type 2 diabetes [18], asthma [19], and rheumatoid arthritis [20], where good health literacy skills are associated with improved condition management. In Parkinson’s disease (PD) low health literacy is associated with greater caregiver burden, reduced access to care resources, reduced engagement of people with PD in healthcare, and delayed end-of-life care planning [21,22,23,24]. However, MND differs from those chronic conditions where good disease management, including self-management, can achieve long periods of disease stability. In MND, there is often rapid progression and relentless change coupled with a decreasing functional independence [4]. It is not known whether the health literacy needs of persons living with MND and carers align with those of people living with other chronic disease. Examining health literacy in MND from the perspective of persons living with MND and carers has the potential to raise awareness of healthcare engagement, and information behaviour research and practice. We chose to use longitudinal qualitative methodology due to the exploratory nature of the study and the need to capture a deep understanding of the lived experience over the disease continuum. We aimed to better understand engagement with information and involvement in healthcare decisions in MND by examining interview data using the construct of health literacy.

## 2. Methods

### 2.1. Context of This Study

This paper is a component of a broader project that explored the lived experience of involvement in healthcare decisions for persons living with MND and carers. The topic guides (Appendix A) relate to this broader study, which explored the decisions that people make, or do not make, about interventions, home modifications, advance care planning, etc., as well as how and with whom they make those decisions. Previous studies provide insight into how persons living with MND and carers experience healthcare decision making [8] and the impact of communication on healthcare involvement [25]. The opportunity to explore the existing data set through the lens of health literacy was not an a priori study aim but was identified during data collection. Thus, questions related to information seeking and receiving were added to the interview guide for the third round of interviews.

### 2.2. Methodology

This exploratory study was underpinned using an interpretivist research paradigm that recognises the importance of understanding the varied perspectives of people in relation to the context and circumstances of their lives [26]. The researchers wished to explore health literacy in MND. Because an extensive body of health literacy literature exists, we used the constructs of a well-established, evidence-based health literacy measuring tool [27] to inform the analytical framework used for deductive analysis (described below). Due to the exploratory and foundational context of the study, we chose to report results with straight descriptions consistent with qualitative descriptive methods [28].

### 2.3. Recruitment

This study recruited persons living with MND and family members (described as carers) from a specialised multidisciplinary clinic in Melbourne, Australia, which annually services approximately 350 people with MND. Potential participants identified at a clinical meeting were provided with a one-page flyer advertising the study by a member of the clinical team. Patients who expressed an interest in participating were provided with a Patient Information and Consent Form (PICF) and contacted by the first author, who answered questions, determined willingness to proceed, and obtained informed consent. The first author was not known to patients or carers at the clinic. Participants were asked to nominate a family member to participate in the carer interviews. Carers were provided with a PICF, and informed consent was obtained by the first author if they agreed to participate. Written consent was obtained on the day of the first interview; participants who were unable to grip a pen, provided verbal or typed consent via their AAC. Three participants declined involving family members, and one carer declined. Carer participants comprised 14 spouses and 1 adult child. One participant responded to the study information poster in the clinic waiting room and emailed the first author directly. Two potential participants that were contacted for the study declined to be involved, citing health issues or insufficient time. Recruitment occurred over a 10-month period.

Participants were sampled to gain a diversity of age, gender, MND phenotype, rate of progression, and verbal and/or non-verbal communication modes (Table 1 and Table 2) consistent with purposeful maximum variation sampling [29]. Only people able to converse in English and provide informed consent (as determined by their treating neurologist) were approached. Patients diagnosed with frontotemporal dementia associated with MND were not studied. Ethical approval was given by the Calvary Health Care Bethlehem Research Ethics & Ethics Committee (reference: 17081701) and the University of Melbourne’s Behavioural and Social Sciences Human Ethics Sub-Committee (reference: 1750285).

### 2.4. Research Sample

A total of 34 participants (19 persons living with MND and 15 carers) were interviewed by the first author between December 2017 and January 2020. Participants were interviewed at baseline (T1) and again at approximately six months (T2) and 14 months (T3). Due to the progressive nature of MND some participants were unable to participate for the duration of the study; three participants died between the T1 and T2 interviews, and a further three were either too unwell to participate or died between T2 and T3. Three participants declined further involvement in the study: one person living with MND and one carer when approached for T2 interview (no reasons provided), and one carer when approached for T3 (citing insufficient time) (Figure 1).

### 2.5. Data Collection

Semi-structured interviews were conducted by the first author. Interviews were audio-recorded, transcribed verbatim, and added to NVivo 12 (QSR International 2019, Burlington, NJ, USA) for collation and management. Interviews lasted on average 50 min, and ranged from 30 to 75 min. Most interviews were conducted in-person in participants’ own homes. A small number of T2 or T3 interviews were conducted by telephone at participants’ requests. Some participant dyads were separately interviewed as intended, and some were jointly interviewed as per their preference. When participants requested to be jointly interviewed, the interviewer commenced the interview acknowledging the risk that this may influence the interview. Participants were offered the opportunity to be separately interviewed if they considered they may not fully disclose their opinions due to the presence of the other; none chose to do so. Sixteen interviews were conducted jointly and forty-nine individually (see Appendix A for interview composition). Only participants who consented to be involved were present for interviews. Participants reliant on AAC were provided with questions in advance and gave preliminary answers over email with focused follow-up questions answered in-person [31].

To preserve anonymity, the gender-neutral singular pronoun (e.g., they or them) was used. Quotes anonymised in this way occasionally read ungrammatically with the singular verb tense; however, it was our preference to retain the rest of the quote as provided. T# indicates the time point from which the quote occurred, and ‘(w)’ indicates the response was written or typed. Due to the potentially sensitive nature of discussing healthcare decisions between persons living with MND and carers, joint interviews quotes are indicated with ‘(j)’.

### 2.6. Data Analysis

The data set included interview transcripts, demographic information, and functional assessment via the ALS Functional Rating Scale (revised) (ALSFRS-R) [30] (Appendix A). Longitudinal data analysis of 60 interview transcripts commenced with data familiarisation, during which all interview transcripts for each participant were consecutively read multiple times. A deliberately diverse selection of participants was chosen to commence the longitudinal analysis. Deductive coding used an a priori concept structure developed from engagement with the health literacy literature [27]. Transcripts were analysed using the framework method [26], together with the application of a trajectory data analysis approach [32]. Both methods use a matrix-style analytic approach. The framework method involves: data familiarisation; initial thematic framework construction; indexing and sorting; reviewing data extracts and revising the thematic framework; data summary and display; and category construction and description [26].

The use of a matrix style approach was helpful to arrange and track longitudinal data, and to retain a temporal structure. Coding of all transcripts was completed in NVivo 12 (QSR International 2019, Burlington, USA) and concurrently summarised into the coding matrix using Excel [26,32]. Data were organised into themes in a matrix; working (or preliminary) subthemes were contained in separate columns, and participants had a row for every interview in which they participated. This structure allowed for both a visual and systematic interrogation of the data to examine how individuals or issues compared at different time points [32]. Paying attention to negative cases ensures that patterns and issues identified during analysis reflected the whole data set [26]. Direct quotes were retained within the matrix to ensure a reliable link to the source data.

### 2.7. Rigour

Rigour of this study is demonstrated by a clear description of study design, data collection and analysis [33] and the researchers’ extended engagement with the data [34]. Three pilot interviews were conducted: a volunteer with a chronic health condition but not MND, a person living with MND regularly involved in medical student training, and a carer of a person living with MND. The latter two interviews were included in the data set. The interview guide was revised following pilot interviews to improve interview coherence and flow [33]. The first author, who conducted all interviews, is trained in qualitative interviewing techniques and is a speech language pathologist with the skills and experience to interact with people with communication and cognitive impairments. The first author attended fortnightly MND multidisciplinary clinical care meetings for six months during the recruitment period. This did not involve patient contact but provided valuable insights regarding the complexity of patient presentations and interdisciplinary care management. This information provided sensitisation to the topic area and helped inform inquiries during interviews. Furthermore, the first author checked the health status of participants prior to interviews to ensure contact was appropriate and sensitively conducted [34]. Emergent concepts were regularly discussed by authors CP and MC, as well as CP, SM and HG to ensure interpretations were defensible, strongly linked to the data source, and clinically relevant. Themes identified during T1 analysis were member checked with nine participants during T3 interviews. Finally, the use of thick description, illustrative quotations, and a comparison to existing literature all support rigour [33]. In this article, the quotes that support the development of themes and subthemes are tabulated underneath the explanatory prose [35,36].

## 3. Results

The results reveal participants’ health literacy behaviours, that is, how they sought, understood, and used healthcare information and engaged in healthcare. The results were organised into five themes: accessing, understanding, and using information, the influence of time, and the influence of HCP (Table 3).

### 3.1. Accessing Information

#### 3.1.1. Seeking Information

More than two-thirds of participants actively engaged in seeking information from a mix of formal and informal sources (Table 4). Almost all information was obtained online from a range of authoritative, evidence-based, alternative/non-traditional websites, medical and scientific journals, and patient experience blogs. Considerable variation was evident amongst participants, from reviewing medical/scientific data and completing a web-based training module aimed at HCPs, through to cursory reading of the MND Association (MNDA) ‘fact sheets’. Three persons living with MND asked paid carers for practical and specific homecare-needs-related information. One participant living with primary lateral sclerosis (PLS) reported that it was difficult to find information about this rarer phenotype.

Most participants who actively sought information were selective in their approach; only a few searched indiscriminately. The most common websites accessed were national and international MND associations and major hospitals. One participant reported they only searched alternative/non-traditional medicine websites. Many participants found value in reading the personal accounts of other persons living with MND, which they accessed via the MNDA newsletter, Facebook groups, YouTube, or patient blogs found on the internet.

#### 3.1.2. Reasons for Not Seeking and Accessing Information

Participants who did not seek information provided mixed reasons. Approximately one-fifth of participants interviewed at T3 reported that seeking information was futile; there is no cure, therefore, no treatment information can be found. Some participants avoided seeking information because they perceived it was too negative. A small group were confident that they were provided with all the information they needed. Carers working full-time missed opportunities to access information when they couldn’t attend clinic or MNDA meetings. Almost half the participants interviewed at T3 reported a reluctance to look to the future which impacted their motivation to obtain information. A small number deliberately avoided accessing information altogether. Participants reliant on non-verbal communication modes reported difficulty accessing information because they could not use the phone or easily access information online due to fatigue.

### 3.2. Understanding Information

Understanding information provided during clinic appointments was commonly supported by reflecting on information and generating questions (Table 5). A small number of participants received a lot of information about prognosis and invasive interventions at their first or second clinic visit, which they found confronting and unexpected. Most of these participants reflected in later interviews they then understood that early information provision was necessary for proactive clinical management, particularly in relation to early gastrostomy placement.

Personal style and family dynamics impacted information needs, which varied between participants and within dyads. For example, two carers reflected they needed information in ‘lay’ terms which differed significantly from their partners who sought ‘high level’ or scientific information. A small group of participants (both persons living with MND and carers) relied on their family to distil information either due to complexity or content. Family members in this group were more likely to have scientific or healthcare backgrounds and were therefore perceived as possessing the expertise to carry out this task.

Communication impairment impacted participants’ ability to ask questions or seek clarification and provide reasoning, which limited the opportunity for a full discussion of issues. Primarily, this communication barrier occurred due to the effort required for persons living with MND to produce speech and the time required when using alternative (non-speech) modes of communication. Persons living with MND reliant on AAC reported that, on occasion, HCPs prematurely anticipated their responses, frustrating their attempts to seek clarification.

### 3.3. Using Information

Information use varied between participants (Table 6). A small number of participants brought evidence-based medical information to appointments for discussion. In contrast, some reported they did not read the provided information. Previous experience with medicolegal paperwork or the aged care system was helpful for using and applying information.

Not all participants reported information was useful, primarily because there was no cure or treatment to consider or because they felt it wasn’t personally relevant at the time. This was reported by participants with varying levels of impairment and disease duration. One person living with MND revealed that they deliberately avoided giving specific information to their family members as a means of protecting them. The influence of patient anecdotes and ‘celebrity faces’ of MND was evident in the way participants discussed and used non-evidence-based information. Patient blogs were useful for participants to contextualise experiences, identify others with unusual symptoms, or to discuss a less common intervention (e.g., tracheostomy).

Communication impairment impacted participants’ ability to ask questions due to the effort required to talk, or the time required to respond in writing or via AAC. Persons living with MND with a communication impairment reported occasions where they provided superficial answers, asked fewer questions, or did not fully express opinions.

### 3.4. The Influence of Time

Many participants in this cohort did not demonstrate significant disease progression over the study period, while others progressed more rapidly and either discontinued the study or died between interview time points. This reduced our ability to identify the impact of functional change on engagement in healthcare longitudinally. The perception of time and disease progression were often indistinguishable in participants’ answers.

Information needs changed, and the understanding of disease management improved over time (Table 7). Some participants described a ‘search, rest, and review’ approach to information seeking. A suspension in accessing information was initiated due to ‘information fatigue’, a lack of new information being found, or emotional reactions. Some, but not all, participants accessed less information as their disease progressed. Generally, participants became more focused on information seeking as disease duration increased.

Reflecting on decision making over time, participants generally believed that decisions became more difficult. This was due to the increased complexity of interventions, the impact of symptom progression, including communication deterioration, and a sense of limited time. Participants with slowly progressing disease thought their decision-making process was easier or more predictable than for those with fast-progressing disease. However, participants with fast-progressing disease did not necessarily believe their process was more complex. In fact, some participants with fast-progressing disease described how disease progression often dictated interventions, thereby effectively removing the need to make a decision.

### 3.5. Influence of Healthcare Professionals

The reported HCP behaviour influenced healthcare engagement in several ways (Table 8). Merit was attributed to open and relaxed consultations, not feeling rushed, and information provision appropriate to needs. Established or long-standing relationships were particularly valued in the case of general practitioners; two participants with dysarthria reported that it was easier to communicate with their GP who knew them prior to development of their speech impairment. Participants who relied on AAC occasionally experienced HCP and were reluctant to use unfamiliar technology, thereby removing the ability to fully engage in their healthcare.

The importance that clinicians made time to personally understand participants was highlighted by a number of participants. A few participants felt misunderstood or unsupported because clinicians did not take time to become acquainted with them or appreciate that they operated as a family unit. HCP skills in facilitating difficult conversations was valued by participants who could not accomplish this outside of clinic. Two participants felt supported by clinicians even though they had refused recommended interventions. The language used by HCP influenced feelings of being understood and supported. The use of qualifiers when HCP provided information regarding future interventions (e.g., ‘if’ or ‘just in case’) fostered engagement; conversely, language that did not reflect participants’ apprehension or concern deterred engagement. All participants who participated in National Disability Insurance Scheme (NDIS) and My Aged Care (MAC) planning [in Australia social care and support is funded through these government agencies] reported that they needed professional support and input to understand and use government social care systems.

## 4. Discussion

This longitudinal exploratory study investigated information engagement and involvement in the healthcare decisions of persons living with MND and carers by examining interview data using the construct of health literacy. The results of this study indicate that persons living with MND and carers sought information from a range of evidence-based and lay sources, which were nuanced depending on their priorities, perceived relevancy, attitudes, and emotional readiness. Information and healthcare engagement is clearly individual and can be impacted by communication and physical impairment. In this study, some participants were not motivated to seek information because they perceived it was not useful due to limited curative treatment information. This contrasts with other reports where most of the surveyed persons living with MND and carers found MND information on the internet or from the local MND Association useful or very useful [10]. Consistent with other studies, we identified that receiving information early or unexpectedly could be emotionally confronting for persons living with MND [8,37], that information preferences sometimes differed between persons living with MND and carers [11,12], and that participants reported benefits from accessing information from peers [13,14]. Only a small number of participants in this study did not access information beyond that provided by the specialist clinic, which differs from the results of Chio et al., (2008), where 45% of persons living with MND and 17% carers did not seek additional information [11]. Prior exposure to the healthcare system was a facilitator of healthcare engagement, which reflects the previously reported needs of carers who required guidance when unfamiliar with the healthcare system [38,39]. Regardless of participants’ engagement with healthcare information or their lived experience with the health system, HCP support (such as that provided by the specialist clinic and the MNDA) was essential for the successful navigation of complex community and social care systems.

Information-seeking behaviour has been primarily explored via questionnaires or surveys [11,14,39]; however, such studies did not examine how persons living with MND and carers understand or use information. Furthermore, the use of semi-structured interviews in this study drew out details that the questionnaires and surveys were unable to, for example, the reason people did not seek information, the importance of the way HCP provide information, and the support that was essential to facilitate the understanding and use of government social care systems. We chose to explore information-seeking behaviour in an existing data set, using the construct of health literacy to refine understanding in this area. As this analysis was retrospective, we did not explore behaviour change. To our knowledge, this is the first study to explore health literacy in detail in people affected by MND.

Functional communication and cognitive skills are central to health literacy behaviour and skills. Despite the high prevalence of these deficits in MND, none of the information-seeking literature (referenced above) reported the influence of communication or cognitive impairment on information seeking. The results from this study clearly indicate the impact that communication impairment has on using and understanding information, and involvement in healthcare decisions. According to participants, communicative interactions with HCPs were sometimes limited or superficial due to the effort to speak or to use AAC. For AAC users, successful communication exchange required HCP to provide extra time and support. Engagement in healthcare was facilitated by HCP accommodating communication impairments and recognising individual information preferences which builds on extant literature exploring patient and carer needs [16,25,39]. Clinicians may need to accommodate or adjust their usual practice to optimise engagement of persons living with MND and carers in healthcare decisions [25]. Constrained or inflexible health system processes have the potential to limit patient engagement in their healthcare [40]. An improved understanding of the person (i.e., both persons living with MND and carers) and the support they need, allows information to be tailored in amount, complexity, and delivery method. Stronger therapeutic relationships may be established if HCP identify personal attitudes and preferences early, drawing out the emotional and psychological needs of persons living with MND and carers [40]. The high prevalence of communication or cognition impairment, disease heterogeneity, as well as uncertain prognosis for some persons living with MND requires an acknowledgement of communication needs and health literacy needs to enhance person-centred care.

## 5. Implications

Information provided to persons living with MND and carers needs to be nuanced. A one size fits all approach will not meet individuals’ needs. Understanding the needs of the person (i.e., both persons living with MND and carers) and the consequences of their health literacy skills will facilitate person-centred care. Communication and cognitive impairment should be accommodated along with the recognition that the needs and preferences of persons living with MND and carers may be different. Even though we do not have the perspectives of HCP for this study, there are implications for clinicians. Clinicians may wish to consider the communication skills suggested below (Table 9), based on Kissane’s (2010) evidence-based consultation framework, to foster this process [41]. Clinicians could challenge themselves to reflect upon the degree of planning for patients’ communication needs and invite patients’ and carers’ agenda items (point 1), check preferences for information and decision-making styles (point 1), and reinforce the value of shared decision making (point 3). Embracing these communication skills would facilitate a patient- and carer-led consultation. Checking patient and carer/family information preferences and understanding (point 2) reflects recommendations in MND research [6] and may establish emotional readiness to receive information [7,8].

## 6. Limitations

Firstly, HCP were not interviewed for this study; therefore, their perspectives are not represented. Consequently, reports of their behaviour are second-hand from persons living with MND and carers. Participants presenting with dysarthria, significant fatigue, and/or respiratory impairment often provided short answers or were unable to respond to follow-up questions. We employed strategies such as emailing participants before or after interviews; however, they were not always taken up by participants. High attrition rates and communication difficulties have previously been identified as obstacles to obtaining the views of persons living with MND [42]. However, researchers are encouraged to challenge the notion that only lengthy verbal discourse is ‘quality’ qualitative data [43]. Participants were recruited from a single specialist multidisciplinary MND clinic; therefore, the experiences of people affected by MND who are not managed under this type of care were not recorded. The lack of cultural diversity in the sample limited the ability to explore the influence of culture and language on health literacy for people impacted by MND. Furthermore, existing research indicates that health literacy influences research participation, whereby people with higher levels of education and higher health literacy are more likely to participate in health research [44]; therefore, bias may exist in the sample. For example, people with low literacy levels may have been unintentionally excluded because they did not understand the study promotional material or plain language statement. Lastly (as discussed in the introduction) the information needs of persons living with MND and carers are often different [6]; therefore, interviewing some dyads together may not have drawn out the specific needs of each.

## 7. Conclusions

The findings of this study indicate that persons living with MND and carers are reliant on successful communication to engage fully with health information and decision making. The framework of health literacy allows us to deconstruct the different steps that patients, carers, and health professionals take when making shared decisions and where those processes of accessing, understanding and using information can be strengthened. The results highlight the potential benefits of gaining an early understanding of and accommodating communication needs, personal preferences, and emotional readiness for information for persons living with MND and their carers. Information needs changed and the understanding of disease management increased over time. The results indicate that persons living with MND and carers do not require access to more information but would benefit from the nuanced provision of information.

## Figures and Tables

**Figure 1 healthcare-10-01371-f001:**
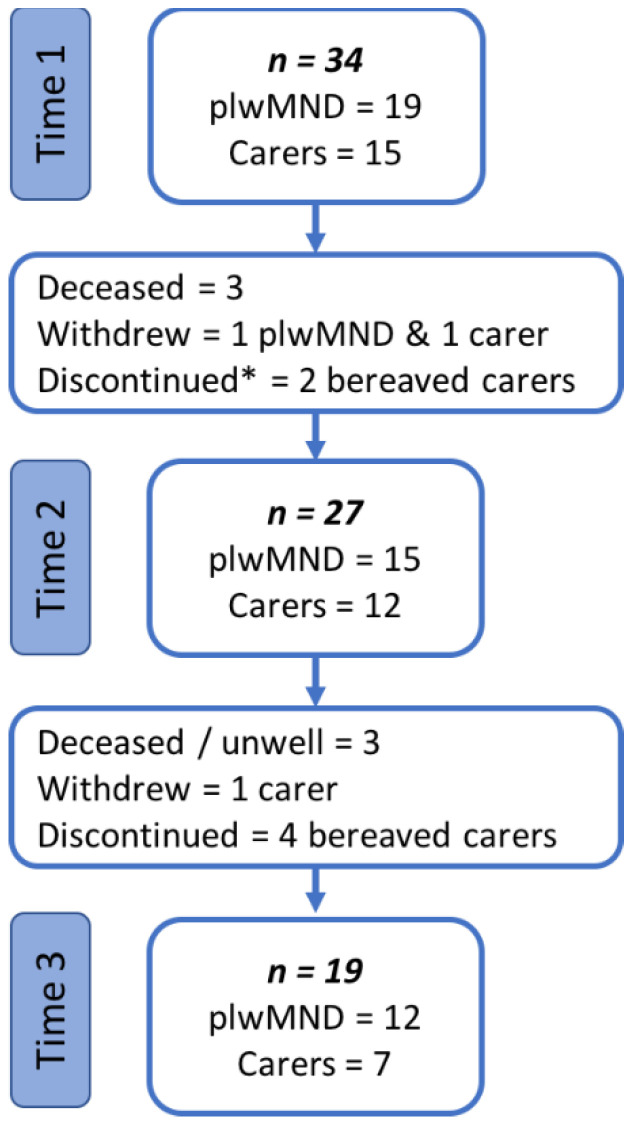
Interview sample at each time point. * One bereaved carer agreed to participate at T2 and is considered to have discontinued from the study at T3.

**Table 1 healthcare-10-01371-t001:** Longitudinal sample characteristics.

Time Point	T1	T2	T3
Persons living with MND (PlwMND) (n)	19	15	12
Females (n)	9	7	6
Age (years)			
Median (Range)	65 (40–79)	67 (41–80)	62 (42–81)
MND Phenotype (n)			
Amyotrophic lateral sclerosis (ALS)	12	9	7
Bulbar onset ALS	4	3	2
Primary lateral sclerosis (PLS)	3	3	3
Years post symptom onset			
ALS, familial ALS, and bulbar onset			
Median (Range)	3.5 (1.1–17.6)	4.1 (1.7–18.3)	3.8 (2.8–6.5)
PLS			
Median (Range)	5.2 (3.8–8.4)	5.7 (4.4–8.9)	7.1 (5.6–10.1)
ALSFRS-R * score (score 48 indicates unimpaired function)			
Total score Median (Range)	33 (10–44)	30 (11–43)	31 (1–41)
ALSFRS-R Subscale score [max 12]: Median (Range)			
Bulbar function	10 (2–12)	10 (2–12)	10 (0–12)
Fine motor function	9 (0–12)	8 (1–11)	7 (0–11)
Gross motor function	8 (0–12)	6 (0–12)	5 (0–12)
Respiratory function	10 (3–12)	9.5 (4–12)	9 (2–12)
Gastrostomy (n)			
PlwMND with gastrostomy (feeding tubes)	3	3	3
PlwMND agreed to gastrostomy but insertion failed	3	0	0
Non-invasive or invasive ventilation (n)	9	7	5
Carers (n)	15	12	7
Females (n)	10	9	5
Age (years)			
Median (Range)	64 (38–73)	60 (39–74)	56 (44–75)
Relationship to plwMND			
Spouse/child	14/1	11/1	7/0

* ALSFRS-R: ALS Functional Rating Scale [30].

**Table 2 healthcare-10-01371-t002:** Participants’ communication function and modes across time points.

Time Point	Verbal Communicator ALSFRS-R Speech Score 4 or 3	Verbal Communicator with Dysarthria ALSFRS-R Speech Score 2	Written Communication Handwriting or Electronic Device ALSFRS-R Speech Score 1 or 0	Communication Device Eye Gaze User ALSFRS-R Speech Score 0
T1 n = 19	n = 14	n = 1	n = 3	n = 1
T2 n = 15	n = 10	n = 3	n = 2	
T3 n = 12	n = 8	n = 1	n = 3	

ALSFRS-R: ALS Functional Rating Scale speech score definition [30]: 4 = normal speech processes; 3 = detectable speech disturbance; 2 = Intelligible with repeating; 1 = speech combined with non-vocal communication; 0 = loss of useful speech.

**Table 3 healthcare-10-01371-t003:** Themes.

Themes	Subthemes
Accessing information	Information seeking behaviour Impact of affect Not seeking information behaviours Barriers to accessing information Avoiding information Communication barriers
Understanding information	Reflection helps understanding Facing conflicting information Information preference Impact of communication impairment
Using information	Active engagement Previous experience facilitates using information Perceptions of usefulness Impression of relevancy Protecting others Influenced by others Communication barriers
The influence of time	Time aids understanding Information needs change Impact of time Perception that slowly progressing disease is easier Disease-dictated decisions making it easier
The influence of healthcare professionals	Relaxed interaction Being understood Accommodation needed for AAC users Need to be understood Impact of language used

**Table 4 healthcare-10-01371-t004:** Accessing Information.

Accessing Information	
Information seeking behaviour	Well um, the people at [Specialist MND Clinic] will tell me. But also my carers have other clients and I ask them questions, like, what happens if I can no longer put myself to bed? What happens? P04 T1
	I always like reading other people’s stories. That’s what I relate to. C15 T1
Impact of affect	It depends on the emotional state I’m in on the day. If I’m in a mood where I do want to know I’ll look [for information]. It just depends on the day. P03 T3 (j)
Not seeking information behaviours	No I don’t [seek information]. Simply because there’s nothing that can be done. So why bother. P14 T2 (j)
	No, I’ve just accepted really that [clinic] seem to be covering everything. So I haven’t looked. P12 T3
	There are some stories where I read about how a patient got diagnosed but then it gets to certain things that scare me. And I don’t want to read anymore. I don’t want to know what’s in the future. P03 T3 (j)
Barriers to accessing information	It’s a bit difficult to get time off work [to attend clinic]. The MND advisor normally comes during the day so I miss [them] and what they talk about. C05 T1
Avoiding information	I told my neurologist, I don’t want to be told yet how long I’ve got. P18 T2 (j)
Communication barriers	I’ve been emailing them [NDIS]. Hard communicating with them. Very tiring, even email and sitting at the computer very tiring. So [child] has been helping. (w) P04 T3

**Table 5 healthcare-10-01371-t005:** Understanding Information.

Understanding Information	
Reflection helps understanding	I feel we’ve come a long way from being absolutely horrified and confronted, to thinking all right, I understand. I wouldn’t say we’ve made a decision, but I think both of us are a lot more informed now. I understand. C15 T1
Facing conflicting information	The [HCP] got involved and said, “no, you’ve got to do it [gastrostomy (PEG)]”. They said that I’d better go and see a [specialist] [who] said, “what are they on about? You don’t need a PEG”. I just don’t know. I’m still confused. I have trouble making up my mind. P01 T2
Information preference	[Spouse] is very interested in the scientific side, I’m very lay and need very basic [information] about what’s happening because I don’t understand all that stuff and [have] a different interest factor. C07 T1
Impact of communication impairment	With the gastroenterologist: probably asked fewer questions than I would have otherwise. In general, I find sometimes I am presented with a list of options to assent/decline, if I want to present my own option I have to stop the speaker and make them wait for me to write the statement. (w) P08 T1
	[Decision making] is much slower, a lot of patience is required to give me time to consider and write my response. (w) P16 T1 (j)

**Table 6 healthcare-10-01371-t006:** Using Information.

Using Information	
Active engagement	I read up about the NIV and how early introduction really seems to improve life span. I was saying [to spouse] you really need to get onto this, sooner rather than later. So, when we went to the clinic we spoke to the respiratory doctor about it. C06 T1
Previous experience facilitates using information	I’ve had to do similar [advance planning] with my dad. And now doing it for myself, yes, it came easier, but it’s also more confronting. Because now you’re doing it for YOU, it’s not for someone else. P03 T2 (j)
Perceptions of usefulness	[The information] is as useful as can be, or as useless as it can be [laughs]. Useful means you can do something with it. When there’s no cure, there’s nothing you can do. P07 T3
Impression of relevancy	I think [specialist clinic] is very diligent. It might not apply to me, but I’ve always said, well it could apply in the future and [specialist clinic] seem to be covering everything. P12 T3
Protecting others	I tend to sort of feel protective of my sisters and towards [spouse] too. My cousin is a doctor so I talk to [them] in more detail. P02 T2
Influenced by others	There’s a few support groups on the internet, most of them are in the UK. Anyhow, just last week someone posted a question about [unusual symptom]. I said “yes, I do”. That’s just one little snippet of information that I think, okay, that’s part of the process”. P09 T3
Communication barriers	I don’t communicate much [with GP]. I use short sentences. It’s a real effort. P14 T2 (j)
	Communicating gets harder, I can still indicate what I want but harder to explain reasoning. (w) P08 T3

**Table 7 healthcare-10-01371-t007:** Influence of Time.

Influence of Time	
Time aids understanding	I can now see why they raised those issues [gastrostomy and advance care planning] in those first appointments. P02 T3
Information needs change	I’m over it [searching for information] I was getting an information overload. I found so much out about it [but] I don’t look for information anymore. P07 T3
Impact of time	I can see, as times goes on, everything is going to get harder, decisions, communicating, everything to do with life will get harder. P07 T3
	[Decisions are] a bit harder because there’s a sense of time running out. (w) P10 T3
Perception that slow progressing disease is easier	I think because my MND is progressing relatively slowly it’s meant that I’ve had time to think about it properly. I’m sure it would be quite different if I had an aggressive form that was changing month to month. P11 T3
Disease-dictated decisions making it easier	Well if you come to the point of no return then it’s easy to, you know, decide. P02 T2
	We’d heard about it [gastrostomy] from the neurologist. [Spouse] was losing weight so it was like, ‘how soon can it be done’. It wasn’t anything that we needed to think about. C06 T1

**Table 8 healthcare-10-01371-t008:** Influence of Healthcare Professionals.

Influence of Healthcare Professionals	
Relaxed interaction	They [neurologist] makes it like a conversation, to explain it to you and it’s understandable. C09 T1
Being understood	The neurologist really understands that [they] is on that [high] level of science. Because initially some people would be really simplifying [information] and I was thinking, “Do not talk like that, [they’re] way past layman’s terms!” It’s important for [spouse] to have someone to discuss this with on [their] level. C07 T3
Accommodation needed for AAC users	Don’t read over my shoulder and try to guess what I am going to say. All too often, people guess incorrectly and record the wrong response or start doing something that I didn’t want. (w) P16 T1 (j)
Need to be understood	They’re [HCP] not interested in anything about me. They’re not holistic in their thinking. And if you offer them any information, they’re not interested either. P05 T3
Impact of language used	The [HCP] said “have you thought anymore about trying the wheelchair?” But I’m not sure I’m ready for that. So [they] said, “look, we might as well put your name down now, because it can take a while and you can try it next time.” It’s good they don’t push [them] too hard. C05 T3 (j)
	Healthcare professionals seem very ‘glib’ about it [gastrostomy]. It’s invasive which is on my mind. They were quite cheery about it. It didn’t address my apprehension. (w) P10 T3
Professional support to navigate social care	I can’t imagine going to NDIS without an MND advisor. We wouldn’t even know where to start. Also the [clinic] put together a report. It’d be a lot harder if you had to pull that information together yourself and work through the system. P11 T2

**Table 9 healthcare-10-01371-t009:** Communication skills to encourage patient- and carer-centred care (abridged and modified from Kissane 2010).

Aim	Communication Skills
Prepare for consultation and set expectations and goals.	Establish communication needs and supports required to facilitate and maximise communicative exchange. Negotiate agenda (i.e., state your agenda items and invite patient/carer agenda items).
	Check patient preferences for information and decision-making style, * including preference for carer/family involvement. Endorse question asking.
Develop an accurate understanding of current disease status, interventions under consideration, and psychosocial needs or concerns.	Check patient and carer/family understanding. Clarify. Invite patient and carer/family concerns.
	Review the information and then summarise. Check patient and carer/family understanding. Endorse question asking. Offer decision delay or abort if patient/carer/family not ready.
Discuss patient, carer/family values and lifestyle factors.	Ask open questions. Clarify. Empathically acknowledge, validate or normalise emotional responses. Reinforce value of shared decision making.
Close the consultation.	Summarise information and discussion. Ask open questions. Affirm value of the discussion. Establish or confirm next steps.

* See [41] for example statements to assist in eliciting this information.

## Data Availability

Due to the nature of this research, participants of this study did not agree for their data to be shared publicly. Additionally, there is risk of participants being re-identified. Therefore, research data are not available.

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
