# Peer review of "Using the Concept of Health Literacy to Understand How People Living with Motor Neurone Disease and Carers Engage in Healthcare: A Longitudinal Qualitative Study"

_healthcare, 2022, doi:10.3390/healthcare10081371_

Round 1

Reviewer 1 Report

In this studyPaynter and colleagues assessed the commitment of patients with motor neuron disease (MND) and their carers to healthcare. The authors claimed that “health literacy” is a powerful concept to be used in understanding such engagement, although lacking specific constructors. Thus, the authors proposed a specifically structured constructor composed of five themes for this goal. While the results are very exciting, this study lays on a qualitative approach, needing therefore a more judicious statistical validation. However, I also understand the complexity of such longitudinal design (about 37% of patients deceased) and the urgency of additional instruments capable to assess and improve patients’ quality of life.

Author Response

Thank you for your feedback and encouragement.

Reviewer 2 Report

I applaud the study investigators on their qualitative investigation of health literacy in people living with MND and their caregivers in Australia. The study provided unique insights relating to how and when information is accessed, and that need to personalise the delivery of this information to optimise uptake in both the patient and caregiver. 

I have only minor grammatical and/or sentence structure comments below, along with one comment relating to the baseline descriptive statistics table. 

Introduction:

1. Please replace the use of the term 'plwMND' throughout the paper with 'persons living with MND'. I understand that the authors quite rightly are empathetic to the cause of not labelling patients simple by their condition (i.e. MNDs), however when the acronym is used, it once again places a label on the patients - 'persons living with MND' should instead be used throughout.

2. Line 53-56, page 2: Gastrostomy improves quality of life for some people [5] although impact on survival remains unclear [4]. Please rephrase for formality and structure to 'For a proportion of people living with MND gastronomy improves quality of life however the impact on survival remains unclear'

3. Multidisciplinary team clinics provide coordinated clinical care and have been shown to enhance survival compared with patients managed solely by general neurology clinics [2].

Please rephrase for formality and structure to 'Coordinated clinical care providers by multidisciplinary team clinics is documented to enhance survival compared with patients managed solely by general neurology clinics [2].'

4. Line 58-60, page 2: Information provision needs to be person-specific as information preferences of plwMND and carers can vary in timing, depth, and topics (see [6] for review). Please add a line summarising briefly the findings of the review referenced. 

5. Line 65 page 2: PlwMND prefer information about disease symptoms, course and prognosis, and potential research options [10-12]. Please do not start a sentence with an acronym - please change in other circumstances if appropriate. 

6. Line 77 - 80, page 2: Health literacy in the MND community has not been as widely explored as it has in other health conditions such as type 2 diabetes [18], asthma [19], and rheumatoid arthritis [20] which demonstrates that good health literacy skills are associated with improved management of those chronic health conditions.

Please rephrase for formality and structure to ' Health literacy in the MND community has not been as widely explored as it has in other chronic health conditions such as type 2 diabetes [18], asthma [19], and rheumatoid arthritis [20] where good health literacy has been associated with improved condition management.

7. Line 85-86, page 2 : In MND, there is often rapid progression and relentless change coupled with decreasing functional independence. Please add a reference. 

8. Line 86 - 87, page 2: Whether the health literacy needs of plwMND and carers align with those of people living with chronic disease is not known. MND is a chronic disease, please rephrase to 'other chronic diseases' for clarity. 

9. Line 90 - 92, page 2: Due to the exploratory nature of the study and to capture a deep understanding of the lived experience over the disease continuum we chose to use longitudinal qualitative methodology

Please rephrase for formality and structure to: 'We chose to use a longitudinal qualitative methodology due to the exploratory nature of the study and the need to capture a deep understanding of the lived experience over the disease continuum'

Methods:

10. Line 98 - 101, page 3: The topic guides (supplementary material) relate to the broader study which explored the decisions that people do, or don’t make, about interventions, home modifications, advance care planning etc, and how and with whom they make those decisions. This sentence is informal, please rephrase to remove 'do or don't' terminology. 

11. Line 101 - 103, page 3: Study results published include how plwMND and carers experience healthcare decision making [8], and the impact of communication on healthcare involvement [25]. 

Please rephrase for formality and structure to 'Previous studies provide insight into how plwMND and carers experience healthcare decision making [8], and the impact of communication on healthcare involvement [25]. '

12. Line 131 - 133, Page 3: Two potential participants contacted declined involvement citing health issues or insufficient time. Recruitment occurred over a 10-month period.

Please rephrase for formality and structure to 'Two potential participants that were contacted for the study declined to be involved citing health issues or insufficient time. Recruitment occurred over a 10-month period'

13. Page 4 -Table 1 - Longitudinal sample characteristics

A. please replace (n=) with (n) usage in table. It is unnecessary

B. I feel as though given the small sample and looking at the position of the mean in relation to the range that non-parametric baseline values should be reported - i.e. Median (range), not the mean - in most cases. This would also allow the inclusion of outliers who have been excluded such as the patient that is 17.6 years post symptom onset. Please correct to median (range) for all relevant categories if not normally distributed. 

14. Some participant dyads were interviewed separately as intended, and some were interviewed jointly as per their preference. Sixteen interviews were conducted jointly and 202 forty-nine individually (see supplementary material for interview composition). This concept needs to be explored further in the limitations section. Currently, the one line at the end of the limitations does not explore the potential influence of the caregiver of patient responses in enough depth. 

Discussion, Implications and Limitations

15. Line 386, page 12: Results show that plwMND and carers sought information from a range of evidence-based and lay sources which was nuanced depending on their priorities, perceived relevancy, attitudes, and emotional readiness. Please rephrase to 'The results of this study indicate that...'

16. Line 407 to 409, page 12: Information seeking behaviour has been explored primarily via questionnaires or surveys [11,14,39] however these studies have not examined how plwMND and carers understand or use information as this study reports. Please rephrase this sentence without the use of '...as this study reports'.

17. Line 416, page 12: . To our knowledge, health literacy has not been explored in detail in MND. 

Please rephrase to: 'To our knowledge, this is the first study to explore health literacy in detail in people living with MND.'

18. Line 444, page 13: Even though we do not have the perspectives of HCP for this study, it does seem there are implications for clinicians. Please be more assertive in your tone for this sentence. 

Line 471 - 475, page 14: Furthermore, existing research indicates health literacy influences research participation [44] therefore bias may exist in the sample. Please define what types of bias this may be. 

-----

Overall the study research methodology and paper composition is sound, and I look forward to seeing this important manuscript published. I hope the authors look to expand their findings to the views of healthcare practitioners in relation to this matter and further refine the clinician's approach to dissemination of health information to those living with MND and their caregivers. 

Author Response

Reviewer 2

Thank you for all your suggestions to improve the readability and quality of this paper, and for your encouraging summary.

Introduction:

  1. Please replace the use of the term 'plwMND' throughout the paper with 'persons living with MND'. I understand that the authors quite rightly are empathetic to the cause of not labelling patients simple by their condition (i.e. MNDs), however when the acronym is used, it once again places a label on the patients - 'persons living with MND' should instead be used throughout.

The abbreviation 'plwMND' has been replaced with 'persons living with MND’ throughout the paper except for tables to accommodate formatting requirements.

  1. Line 53-56, page 2: Gastrostomy improves quality of life for some people [5] although impact on survival remains unclear [4]. Please rephrase for formality and structure to 'For a proportion of people living with MND gastronomy improves quality of life however the impact on survival remains unclear'

 The suggestion has been implemented and the text changed. Line 53-54, page 2

  1. Multidisciplinary team clinics provide coordinated clinical care and have been shown to enhance survival compared with patients managed solely by general neurology clinics[2].

Please rephrase for formality and structure to 'Coordinated clinical care providers by multidisciplinary team clinics is documented to enhance survival compared with patients managed solely by general neurology clinics [2].'

 The suggestion has been implemented and the text changed. Line 55-56, page 2

  1. Line 58-60, page 2: Information provision needs to be person-specific as information preferences of plwMND and carers can vary in timing, depth, and topics (see [6] for review). Please add a line summarising briefly the findings of the review referenced. 

The sentence preceding “(see [6] for review) summarises the findings of the review. The sentence has been revised to remove ambiguity and now reads “Information provision needs to be person-specific as information preferences of plwMND and carers can vary in timing, depth, and topics [6].” Line 61, page 2

  1. Line 65 page 2: PlwMND prefer information about disease symptoms, course and prognosis, and potential research options [10-12]. Please do not start a sentence with an acronym - please change in other circumstances if appropriate. 

This sentence has been revised to start with “Persons living with MND”. The manuscript has been checked and revised where necessary to ensure no other sentences commence with an abbreviation.

  1. Line 77 - 80, page 2: Health literacy in the MND community has not been as widely explored as it has in other health conditions such as type 2 diabetes [18], asthma [19], and rheumatoid arthritis [20] which demonstrates that good health literacy skills are associated with improved management of those chronic health conditions.

Please rephrase for formality and structure to ' Health literacy in the MND community has not been as widely explored as it has in other chronic health conditions such as type 2 diabetes [18], asthma [19], and rheumatoid arthritis [20] where good health literacy has been associated with improved condition management.

 The suggestion has been implemented and the text changed. Line 81-82 Page 2

  1. Line 85-86, page 2 : In MND, there is often rapid progression and relentless change coupled with decreasing functional independence. Please add a reference. 

 Thank you for noticing this omission. A reference (Keirnan, et al. 2011) has been inserted.

  1. Line 86 - 87, page 2: Whether the health literacy needs of plwMND and carers align with those of people living with chronic disease is not known. MNDis achronic disease, please rephrase to 'other chronic diseases' for clarity. 

Thank you for noticing this error. The sentence has been corrected and now reads “It is not known whether the health literacy needs of persons living with MND and carers align with those of people living with other chronic disease.” Line 89-90 Page 2

  1. Line 90 - 92, page 2: Due to the exploratory nature of the study and to capture a deep understanding of the lived experience over the disease continuum we chose to use longitudinal qualitative methodology

Please rephrase for formality and structure to: 'We chose to use a longitudinal qualitative methodology due to the exploratory nature of the study and the need to capture a deep understanding of the lived experience over the disease continuum'

 The suggestion has been implemented and the text changed. Line 93-96 Page 2.  

Methods:

  1. Line 98 - 101, page 3: The topic guides (supplementary material) relate to the broader study which explored the decisions that people do, or don’t make, about interventions, home modifications, advance care planning etc, and how and with whom they make those decisions. This sentence is informal, please rephrase to remove 'do or don't' terminology. 

 The sentence has been revised and now reads “The topic guides (supplementary material) relate to the broader study which explored the decisions that people make, or do not make, about interventions….” Line 104 Page 3

  1. Line 101 - 103, page 3: Study results published include how plwMND and carers experience healthcare decision making [8], and the impact of communication on healthcare involvement [25]. 

Please rephrase for formality and structure to 'Previous studies provide insight into how plwMND and carers experience healthcare decision making [8], and the impact of communication on healthcare involvement [25]. '

 The suggestion has been implemented and the text changed. Line 105-106 Page 3 

  1. Line 131 - 133, Page 3: Two potential participants contacted declined involvement citing health issues or insufficient time. Recruitment occurred over a 10-month period.

Please rephrase for formality and structure to 'Two potential participants that were contacted for the study declined to be involved citing health issues or insufficient time. Recruitment occurred over a 10-month period'

 The suggestion has been implemented and the text changed. Line 137 Page 3

  1. Page 4 -Table 1 - Longitudinal sample characteristics
  2. please replace (n=) with (n) usage in table. It is unnecessary

The table has been revised and only includes “(n)”

  1. I feel as though given the small sample and looking at the position of the mean in relation to the range that non-parametric baseline values should be reported - i.e. Median (range), not the mean - in most cases. This would also allow the inclusion of outliers who have been excluded such as the patient that is 17.6 years post symptom onset. Please correct to median (range) for all relevant categories if not normally distributed. 

Thank you for your helpful suggestion. Median values have replaced mean values, and the line explaining an outlier was excluded has been removed as it is no longer appropriate.

  1. Some participant dyads were interviewed separately as intended, and some were interviewed jointly as per their preference. Sixteen interviews were conducted jointly and forty-nine individually (see supplementary material for interview composition).This concept needs to be explored further in the limitations section. Currently, the one line at the end of the limitations does not explore the potential influence of the caregiver of patient responses in enough depth. 

 Thank you for your comments regarding this issue. Whilst there is potential for either participant to influence the other, there was also sometimes benefit of having the caregiver present to assist with understanding the person living with MND. In response to your comment, further detail has been added section 2.5 Data Collection to report the effort the authors undertook to reduce the risk of influence. The following sentences have been added:

“When participants requested to be interviewed jointly, the interviewer commenced the interview acknowledging the risk that this may influence the interview. Participants were offered the opportunity to be interviewed separately if they considered they may not fully disclose their opinions due to the presence of the other; none chose to do so.” Line 277-281 Page 6

Discussion, Implications and Limitations

  1. Line 386, page 12: Results show that plwMND and carers sought information from a range of evidence-based and lay sources which was nuanced depending on their priorities, perceived relevancy, attitudes, and emotional readiness.Please rephrase to 'The results of this study indicate that...'

 The suggestion has been implemented and the text changed.Line 469 Page 11

  1. Line 407 to 409, page 12: Information seeking behaviour has been explored primarily via questionnaires or surveys [11,14,39] however these studies have not examined how plwMND and carers understand or use information as this study reports. Please rephrase this sentence without the use of '...as this study reports'.

 The suggestion has been implemented and the text changed. Line 493 Page 12  

  1. Line 416, page 12: . To our knowledge, health literacy has not been explored in detail in MND. 

Please rephrase to: 'To our knowledge, this is the first study to explore health literacy in detail in people living with MND.'

This sentence has been revised as per the reviewer’s suggestion. It reads “To our knowledge, this is the first study to explore health literacy in detail in people affected by MND”. The usage of ‘people affected by MND’ is used to encompass the views of carers. Line 501 Page 13

  1. Line 444, page 13: Even though we do not have the perspectives of HCP for this study, it does seem there are implications for clinicians.Please be more assertive in your tone for this sentence. 

This sentence has been revised to read “Even though we do not have the perspectives of HCP for this study, there are implications for clinicians”.  Line 531 Page 13

Line 471 - 475, page 14: Furthermore, existing research indicates health literacy influences research participation [44] therefore bias may exist in the sample. Please define what types of bias this may be. 

This paragraph has been revised to read: “Furthermore, existing research indicates health literacy influences research participation whereby people with higher levels of education and higher health literacy are more likely to participate in health research [44] therefore bias may exist in the sample.”  Line 563 Page 14